# Protein Phosphatase Ppz1 Is Not Regulated by a Hal3-Like Protein in Plant Pathogen *Ustilago maydis*

**DOI:** 10.3390/ijms20153817

**Published:** 2019-08-05

**Authors:** Chunyi Zhang, Antonio de la Torre, José Pérez-Martín, Joaquín Ariño

**Affiliations:** 1Departament de Bioquímica i Biologia Molecular and Institut de Biotecnologia i Biomedicina, Universitat Autònoma de Barcelona, Cerdanyola del Vallès, 08193 Barcelona, Spain; 2Instituto de Biología Funcional y Genómica (CSIC), 37007 Salamanca, Spain

**Keywords:** protein phosphorylation, protein phosphatase, CoA biosynthesis, *Ustilago maydis*, fungi

## Abstract

Ppz enzymes are type-1 related Ser/Thr protein phosphatases that are restricted to fungi. In *S. cerevisiae* and other fungi, Ppz1 is involved in cation homeostasis and is regulated by two structurally-related inhibitory subunits, Hal3 and Vhs3, with Hal3 being the most physiologically relevant. Remarkably, Hal3 and Vhs3 have moonlighting properties, as they participate in an atypical heterotrimeric phosphopantothenoyl cysteine decarboxylase (PPCDC), a key enzyme for Coenzyme A biosynthesis. Here we identify and functionally characterize Ppz1 phosphatase (UmPpz1) and its presumed regulatory subunit (UmHal3) in the plant pathogen fungus *Ustilago maydis*. UmPpz1 is not an essential protein in *U. maydis* and, although possibly related to the cell wall integrity pathway, is not involved in monovalent cation homeostasis. The expression of UmPpz1 in *S. cerevisiae* Ppz1-deficient cells partially mimics the functions of the endogenous enzyme. In contrast to what was found in *C. albicans* and *A. fumigatus*, UmPpz1 is not a virulence determinant. UmHal3, an unusually large protein, is the only functional PPCDC in *U. maydis* and, therefore, an essential protein. However, when overexpressed in *U. maydis* or *S. cerevisiae*, UmHal3 does not reproduce Ppz1-inhibitory phenotypes. Indeed, UmHal3 does not inhibit UmPpz1 in vitro (although ScHal3 does). Therefore, UmHal3 might not be a moonlighting protein.

## 1. Introduction

Ppz1 defines a subfamily of Ser/Thr protein phosphatases (PPases) which are structurally related to type-1 (PP1c) enzymes. In contrast to the ubiquitous PP1c, Ppz1 phosphatases are restricted to fungi [1,2]. The enzyme has been extensively characterized in *S. cerevisiae* as a 692-residue protein whose C-terminal domain is similar to the sole PP1c (Glc7), and is preceded by a long N-terminal segment (~350 residues) unrelated to other proteins [3,4]. ScPpz1 plays a key role in monovalent cation homeostasis through regulation of the function of the high-affinity Trk transporters and the expression of the *ENA1* gene, encoding a Na^+^-ATPase [5,6,7,8]. *S. cerevisiae ppz1* mutants are hypertolerant to Na^+^ and Li^+^ cations and, likely due to the increased influx of potassium, they are sensitive to agents affecting cell wall remodeling, such as caffeine or calcofluor white [4,9,10]. High levels of Ppz1 are toxic in *S. cerevisiae*, resulting in slow growth and cell cycle blockage at the G_1_–S transition [9,11,12]. ScPpz1 is regulated by its inhibitory subunits Hal3 and Vhs3, which interact with a 1:1 stoichiometry to the catalytic domain of ScPpz1 [9,13,14,15,16]. Hal3 is more relevant in vivo than Vhs3 and, in contrast to *ppz1* strains, *hal3* mutants are sensitive to Na^+^ and Li^+^, whereas its overexpression confers tolerance to high concentrations of these cations [13].

Homologs of ScPpz1 have been identified in other fungi, such as *Schizosaccharomyces pombe*, *Neurospora crassa* and *Debaryomyces hansenii* [17,18,19], although they have been characterized in less detail. Recent work has pointed out a role in virulence for Ppz1 enzymes in certain pathogenic fungi, such as *Candida albicans* and *Aspergillus fumigatus* [20,21,22,23], although this does not seem to be the case in *Cryptococcus neoformans* [24]. From a structural point of view, Ppz enzymes display a highly conserved catalytic C-terminal half, but their N-terminal half is usually much shorter than that of *S. cerevisiae* and, although their similarity in sequence is relatively low, in all cases investigated so far, the N-terminus includes a likely myristoylable Gly2. In *S. cerevisiae*, change of Gly-2 to Ala results in a lack of myristoylation and the loss of complementation of salt tolerance [25]. As a rule, the heterologous expression of Ppz enzymes in *S. cerevisiae* can fulfill, at least partially, many of the functions of the native enzyme.

The observation that, contrary to Ppz enzymes, orthologues of *HAL3* are widely distributed in both prokaryotic and eukaryotic organisms is explained by the fact that ScHal3 (and ScVhs3) are moonlighting proteins that, in addition to regulate Ppz1, are involved in the biosynthesis of coenzyme A by contributing to the generation of an unusual heterotrimeric phosphopantothenoylcysteine decarboxylase (PPCDC) enzyme [26]. In most organisms, PPCDC is a homotrimer in which the three equivalent catalytic sites are formed at the interface of the monomers, and involve specific His, Asn (within a conserved PXMNXXMW motif) and Cys residues (i.e., His^90^, Asn^142^, and Cys^175^ in the *Arabidopsis thaliana* Hal3a protein, see Figure 1). In contrast, in *S. cerevisiae* PPCDC is a heterotrimer made up of an essential Cab3 subunit and two Hal3 or Vhs3 subunits (or one of each). Cab3 participates in this complex by providing the catalytic Cys^478^ and Asn^443^ residues (absent in Hal3 and Vhs3), whereas Hal3 or Vhs3 contribute with the catalytic His (His^378^ and His^459^, respectively, see Figure 1 and Appendix A) [26]. *S. cerevisiae* Hal3 also differs from *A. thaliana* AtHal3 PPCDC (the prototype for PPCDC in most organisms) in that the conserved PPCDC central core is flanked by a relatively long N-terminal extension and by an unusually acidic C-terminal tail (Figure 1). Contrary to the relative homogeneity of fungal Ppz enzymes, Hal3-like proteins are much more varied. Thus, the *C. albicans* genome encodes two proteins whose structures are reminiscent of those of ScCab3 and ScHal3 (orf19.3260 and orf19.7378, respectively) [27]. They can replace the corresponding *S. cerevisiae* protein when tested for their ability to build up an heteromeric PPCDC, but only CaCab3 (and not CaHal3) exhibits Ppz1 phosphatase regulatory potential in vivo [27]. Even more striking is the case of the fission yeast *S. pombe*, where SpHal3 was shown to be the result of a gene fusion event composed of an N-terminal PPCDC core (including a short acidic tail), followed by a functional C-terminal thymidylate synthase [28]. Interestingly, SpHal3 was a functional PPCDC, but its capacity to inhibit ScPpz1 or its *S. pombe* homolog (Pzh1) in vitro was limited. Very recently, the Ppz/Hal3 system has also been investigated in *C. neoformans* [24]. This fungus encodes two Hal3-like proteins (CnHal3a and CnHal3b) which essentially consist of a PPCDC core and a non-acidic C-terminal tail. Interestingly, whereas both are functional PPCDC enzymes, none of them appears to regulate Ppz enzymes in vivo or in vitro. In conclusion, fungal Hal3-like proteins are very diverse and, whereas their PPCDC capacity can be reasonably inferred from their structure, in the absence of experimental data, it is not feasible to predict their ability to regulate Ppz phosphatases.

*Ustilago maydis*, the causal agent of corn smut disease, is a plant pathogenic fungus that causes tumors on all aerial parts of its host plant maize (*Zea mays*). Together with the human pathogen *C. neoformans*, it serves as model system of choice for pathogenic basidiomycetes [29]. The fungus has a dimorphic life cycle with a yeast-like saprophytic phase switching to filamentous, pathogenic growth upon hyphal fusion. The amenability for reverse-genetics, as well as its intricate life style, is very connected with cell cycle regulation, making *U. maydis* a model system for both fungal cell biology as well as the study of biotrophic plant interaction [30]. Analysis of the *U. maydis* genome revealed a single gene (umag_04827, named here UmPpz1) which likely encodes a Ppz1 phosphatase, and a second gene (umag_05916) which encodes a sequence that is compatible with a putative PPCDC, denoted here as UmHal3. Remarkably, the latter was an unusually large protein (1430-residue) in which the PPCDC domain is restricted to the last 240 amino acids and is not followed by an acidic C-terminal tail. The fact that Ppz phosphatases have never been investigated in plant fungal pathogens and the abnormal structure of the sole Hal3-like protein found in *Ustilago* (Figure 1 and Appendix A) prompted us to explore the nature of the Ppz/Hal3 system in this fungus. Our investigations reveal that UmPpz1 is not an essential protein in *U. maydis*, and that it might be functionally related to the cell wall integrity pathway, but that it is not involved in maintenance of monovalent cation homeostasis. UmHal3 does not inhibit in vitro UmPpz1 and does not behave as an in vivo regulator of the phosphatase, but it is the only functional PPCDC in *U. maydis* and an essential protein. Therefore, UmHal3 might lack moonlighting properties. In contrast to what has been reported for the phosphatase in *C. albicans* and *A. fumigatus*, UmPpz1 is not a virulence determinant.

## 2. Results

### 2.1. Identification of Ustilago maydis Ppz1 and Hal3-Like Proteins

Blastp analysis of fungal databases using the *S. cerevisiae* Ppz1, Hal3 and Cab3 protein sequences revealed a putative ortholog for *PPZ1* in *Ustilago maydis*, encoding the hypothetical protein UMAG_04827.1, and a single putative ortholog for Hal3 and Cab3 (hypothetical protein UMAG_05916.1). As shown in Figure 1, UMAG_04827.1 (from now on UmPpz1) is a 497-residue protein that shows an overall 38.2% identity (46.5% similarity) with ScPpz1. The C-terminal section of UmPpz1, containing the characteristic traits of a Ser/Thr protein phosphatase, shows high identity with the C-terminal phosphatase domain of ScPpz1 (67.9%, 77.6% similarity from residue 151 to the C-terminal residue). Therefore, the N-terminal extension of UmPpz1 is about half the size of ScPpz1 (Figure 1A). Inspection of the very N-terminal region of this ORF shows that the initiating Met is not followed by a Gly, but by a Phe, indicating that this protein cannot by N-terminally myristolylated (Figure 1A).

The UMAG_05916.1 protein (from now on, UmHal3) is a very large polypeptide of 1430 residues. The last 240 amino acids show significant identity with known PPCDCs, such as AtHal3a (34.2% identity, 48.9% similarity), as well as with the PPCDC core of both ScHal3 (24.1%/39.7%) and ScCab3 (26.8%/40.5%), although the Ustilago protein lacks the characteristic C-terminal acidic tail found in *S. cerevisiae* (Figure 1B) and *C. albicans*. Therefore, UmHal3 has a very long N-terminal extension whose similarity to the one present in Hal3 or Cab3 is only marginal (Appendix A). To note, UmHal3 presents at positions 777–894 a Vps9-like sequence. In fact, the entire Vps9 can be aligned with UmHal3 (from residue 570 to 1096), albeit with relatively low identity (19.4%, 35.5% similarity). In *S. cerevisiae*, Vps9 acts as a guanine nucleotide exchange factor (GEF) for Vps21, and is needed for the transport of proteins from biosynthetic and endocytic pathways into the vacuole.

### 2.2. Expression and Functional Characterization of UmPpz1 in S. cerevisiae

To this end, the UmPpz1 open reading frame (ORF) was expressed from the *ScPPZ1* gene promoter in both centromeric (low-copy number) and episomal (high-copy) vectors. As shown in Figure 2A, expression of UmPpz1 from low or high-copy number vectors was unable to decrease the abnormally high tolerance of the *ppz1* mutant to LiCl. The mutation of *PPZ1* yields *S. cerevisiae* cells which are hypersensitive to caffeine; the sensitivity to the drug is normalized when a centromeric copy of ScPpz1 is reintroduced in the cells. Transformation of *ppz1* cells with centromeric UmPpz1 increased the tolerance to caffeine only marginally, but high-copy expression of UmPpz1 restored wild-type tolerance (Figure 2A). The fact that cells expressing UmPpz1 from the episomal vector grow better in the presence of caffeine than cells expressing ScPpz1 can be explained by previous observations that high-copy expression of ScPpz1 from its own promoter has a negative effect on cell growth [9]. A modest beneficial effect of the expression of UmPpz1 was also observed when the experiment was carried out in a *slt2* mutant, which is defective in cell-wall remodeling and highly sensitive to caffeine.

Although a Gly is not found as a second residue at the defined N-terminal region of UmPpz1, residue number nine is a Met followed by a Gly (Figure 1A). In spite of the fact that computational analysis of the residues following this Gly does not predict this residue being myristoylated [31], we considered the possibility that the actual protein may initiate at this second Met and, therefore, we expressed a C-terminally HA-tagged shorter version (UmPpz1-s-HA) in *S. cerevisiae*. However, as shown in Appendix A, this version not only failed to decrease the tolerance of a *ppz1* mutant to Li^+^ cations but, in contrast to UmPpz1-HA, was totally ineffective in restoring normal tolerance to caffeine. Immunoblot experiments failed to detect tagged UmPpz1-s-HA, suggesting that the protein was very poorly expressed or readily degraded (not shown). Therefore, the first few amino acids in UmPpz1 might be important for protein stability.

As mentioned, strong overexpression of Ppz1 has harmful effects on *S. cerevisiae* cell growth. For instance, the expression of the phosphatase in a pYES2 vector from the powerful *GAL1-10* promoter has been shown to fully block cell proliferation ([25] and Figure 2B). However, expression of UmPpz1 under the same conditions did not affect growth (Figure 2B). The absence of effect was not due to insufficient expression levels, since immunoblot experiments showed that HA-tagged UmPpz1 accumulates even more abundantly than ScPpz1 (Figure 2C). Therefore, UmPpz1 does not reproduce the toxic effects of ScPpz1. The same result was obtained by overexpression of UmPpz1-s (not shown).

### 2.3. Functional Characterization of UmHal3 in S. cerevisiae

Expression of UmHal3 in *S. cerevisiae* was accomplished by cloning the entire ORF in the high-copy plasmid pWS93. In this way, the protein is expressed from the *ADH1* promoter as a N-terminally 3xHA epitope-tagged polypeptide. In addition, we cloned in the same vector the C-terminal region of UmHal3, corresponding to the PPCDC domain (UmPD region) and this same region followed by the highly acidic C-terminal tail found in ScHal3 (UmPD_ScCtD) (Figure 1 and Appendix A). As shown in Figure 3A, expression of UmHal3 or its PD region in a *hal3* strain did not improve tolerance to LiCl or NaCl, which is a typical effect of ScHal3 overexpression. Interestingly, expression of UmPD_ScCt increased tolerance to LiCl somewhat. In contrast, neither version restored wild-type tolerance to the toxic cation spermine or to caffeine. In addition, overexpression of any of the UmHal3 variants had a negative effect on the growth of a *slt2* mutant (Appendix A). We considered whether the virtual lack of effects derived from the introduction in *S. cerevisiae* of the UmHal3 protein and its variants could be attributed to defective expression. To test this, we monitored the expression levels of these proteins taking advantage of the N-terminal 3xHA tag. However, as observed in Figure 3B, all UmHal3 variants, including the very large, full-length UmHal3, were expressed at considerable levels. Therefore, the absence of phenotypic effects must be attributed to the inherent properties of these proteins. Because the phenotypes tested are functionally related to the role of ScHal3 as inhibitor of Ppz1, our results suggest that UmHal3 is not able to inhibit Ppz1 in *S. cerevisiae*.

### 2.4. Characterization of Recombinant UmPpz1 and UmHal3 Proteins

To this end, UmPpz1 was expressed in *E. coli* as a fusion protein with glutathione S-transferase (GST), purified by glutathione agarose affinity chromatography and the GST moiety removed by treatment with PreScission protease. The recombinant enzyme (as well as UmPpz1-s) showed phosphatase activity against the synthetic substrate pNPP which was similar to that observed for ScPpz1 or its C-terminal catalytic domain (ScPpz1-Cter). Our attempts to express and purify significant amounts of full-length UmHal3 in *E. coli* were unsuccessful, possibly due to the large size of the protein. In contrast, we were able to purify both the UmPD and UmPD_ScCt polypeptides. We first set an assay to test the inhibitory properties of the Ustilago Hal3 proteins. As presented in Figure 4, neither UmPD nor UmPD_ScCt were able to inhibit recombinant ScPpz1, ScPpz1-Cter, or even UmPpz1. In contrast, ScHal3 was fully effective as an in vitro inhibitor of UmPpz1. This implies that UmPpz1 retains the structural features required to be inhibited by functional Hal3 proteins, and suggests that at least the UmPD and UmPD_ScCt variants of UmHal3 are devoid of Ppz1 inhibitory properties.

The inability of UmHal3 proteins to inhibit ScPpz1 or UmPpz1 could be due to the inability of these proteins to effectively interact with the phosphatases. To test this, we set a pull-down assay using as baits the recombinant GST-tagged phosphatases bound to the glutathione resin that were incubated with *S. cerevisiae* extracts containing equivalent amounts of HA-tagged ScHal3, UmPD or UmPD_ScCt (Appendix A). As shown in Figure 5A, ScHal3 effectively interacts with UmPpz1, whereas full-length UmHal3 and UmPD are barely visible (see Appendix A for an overloaded version of the experiment) and UmPD_ScCt cannot be detected. On the other hand (Figure 5B), full-length UmHal3 weakly interacts with both ScPPz1 and ScPpz1-Cter. Weak signals can be also observed for UmPD and UmPD_ScCt when ScPpz1 is used as bait, but they are not detected when the interaction with ScPpz1-Cter is tested (even if these lanes are overloaded, as shown in Figure 5B). Collectively, our results indicate that ScHal3 can effectively bind to and inhibit UmPpz1, and that UmHal3 can (although very weakly) interact with ScPpz1 and ScPpz1-Cter, but that it is completely unable to inhibit their phosphatase activity.

### 2.5. UmHal3 Is a Functional PPCDC Enzyme

UV-visible scanning of purified recombinant UmPD preparations showed peaks at 382 and 452 nm, characteristic of the presence of an oxidized flavin, which is a cofactor involved in the PPCDC catalytic process (Appendix A). In addition, UmHal3 contains all the structural determinants necessary for PPCDC activity (i.e., catalytic His, Asn and Cys residues, see Figure 1 and Appendix A). To confirm the possible PPCDC function of UmHal3, we set out two functional tests. First, we transformed the *E. coli* strain BW369 with pGEX-derived vectors containing UmHal3, UmPD and UmPD_ScCt. This strain carries the temperature-sensitive *dfp-707^ts^* mutation that affects PPCDC function; consequently, cells cannot grow at 37 °C [32,33]. In parallel, this strain was transformed with AtHal3 (positive control) or ScHal3 (negative control). As shown in Figure 6A, all three versions of UmHal3 were able to restore growth of the BW369 strain at 37 °C, suggesting that all three are able to provide PPCDC activity (even if UmHal3 is presumably expressed at very low levels, see above). In a second test, UmHal3 derivatives were introduced in diploid *S. cerevisiae* strains heterozygous for the *cab3 or* the *hal3 vhs3* mutations, and the ability of the *Ustilago* proteins to rescue the deletion mutants was monitored by random spore analysis and/or tetrad analysis. As observed in Figure 6B, the expression of all three Ustilago variants allowed growth of the mutant haploid strains, as deduced from the germination of all four spores in the tetrads. Therefore, full-length UmHal3 can substitute either Cab3 and Hal3 (and Vhs3) in *S. cerevisiae* and provide PPCDC activity. As expected, this property can be assigned to the most C-terminal section of the protein.

### 2.6. Phenotypic Characterization of UmPpz1 and UmHal3 Deletion Mutants

*U. maydis* clones carrying a disruption of the gene encoding UmPpz1 were obtained as described in Materials and Methods by replacing one allele in the diploid FBD11 strain with a null allele carrying a hygromycin resistance (HygR) marker. After sporulation, we analyzed the meiotic progeny of these strains; roughly half of the cells in the progeny of the diploid strain carrying the *ppz1*Δ allele were Hyg resistant, indicating that the gene was not essential. We then tested the mutant strain for phenotypes that are characteristic of the deletion of *PPZ1* in budding yeast. As shown in Figure 7, the *U. maydis* mutant (in contrast to the *S. cerevisiae ppz1* mutant) did not display altered tolerance to NaCl or LiCl, and was not sensitive to heat or oxidative stress (H_2_O_2_). *U. maydis* cells deficient in UmPpz1 also displayed normal tolerance to calcium ions or chlorpromazine, an activator of Cell Wall Integrity (CWI) pathway in this fungus [34] (not shown). In contrast, the mutant was more sensitive to caffeine and calcofluor white, two compounds known to interfere with normal cell wall construction. As shown above, such sensitivity is characteristic of *S. cerevisiae ppz1* mutants.

In budding yeast, overexpression of Hal3 yields phenotypes similar to those observed for the *ppz1* mutant (i.e., sensitivity to caffeine or calcofluor white). To test whether this was the case for UmHal3, the entire protein, the C-terminal domain (residues 1187 to the stop codon, equivalent to the PPCDC domain) and the large N-terminal fragment were cloned upstream the *cgr1* promoter, which is activated by arabinose. As shown in Figure 8, none of these proteins resulted in altered tolerance to caffeine or calcofluor white when overexpressed, suggesting that they did not have the expected effect on UmPpz1 function. The fact that UmHal3 is the only protein with a likely PPCDC domain in Ustilago led us to predict that the gene might be essential. To test this, we disrupted one allele in the diploid FBD11 strain with the HygR marker. After analysis of 50 spores, we did not find any Hyg resistant cells among the haploid progeny. These results confirmed that *HAL3* is an essential gene in *U. maydis*.

### 2.7. Ppz1 Is Dispensable for Virulence in U. maydis

Since *U. maydis* is a plant pathogenic fungus, we were curious about the effects of a lack of Ppz1 function in the ability of this fungus to infect plants. The virulence of *U. maydis* is dependent on mating; therefore, first we examined the behavior of haploid strains deleted for *ppz1* with respect to the ability to mate and to produce dikaryotic filament, which can be observed as a white-appearing mycelial growth (fuzzy phenotype) [35]. We crossed compatible wild-type and *ppz1*Δ strains on charcoal-containing plates. Wild-type crosses led to a typical fuzzy colony appearance. In contrast, *ppz1*Δ strains mutants were slightly attenuated in filament formation (Figure 9A).

Following these results, we analyzed the ability of *ppz1*Δ strains to infect and cause symptoms in plants. We inoculated by stem injection maize plants with mixtures of wild type and mutant strains. *U. maydis* infection of maize results in anthocyanin pigment production by the plant and the formation of tumors that are filled with proliferating fungal cells that eventually differentiate into black teliospores [36]. We found that the *ppz1*Δ strains were able to induce pigmentation and tumors on the stems and leaves of infected plants at similar ratio as wild-type (Figure 9B), indicating a minor, if any, role of Ppz1 during the infection process.

## 3. Discussion

The expression of UmPpz1 in *S. cerevisiae* shows that this phosphatase could partially replace endogenous Ppz1 in *S. cerevisiae*, suggesting that the *Ustilago* enzyme has the capacity to impact on specific intracellular targets of the *Saccharomyces* phosphatase. Thus, the introduction of UmPpz1 in *S. cerevisiae ppz1* or *slt2* mutants improves tolerance to caffeine. However, and in contrast to what has been found for Ppz enzymes from other organisms [17,18,20,22,24,28] UmPpz1 does not complement at all the *ppz1* mutation with respect to tolerance to Li^+^ cations. It could be hypothesized that the lack of the arginine-rich conserved motif (SxRSxRxxS) in the N-terminal half of UmPpz1, which was found to be important for cation tolerance in *D. hansenii* [19], could be the reason for such failure. However, this motif is also absent in the N-terminal region of CnPpz1, and expression of this enzyme in *S. cerevisiae* partially normalizes Li^+^ tolerance [24]. Besides, a CaPpz1 mutated version lacking this conserved motif retains the ability to restore normal Li^+^ tolerance in vivo [37]. A second commonly lacking structural feature in UmPpz1 is the N-terminal myristoylable Gly. Previous work has shown that mutation of this Gly to Ala in ScPpz1 greatly affects the role of the Ppz1 on Li^+^ tolerance, but does not influence the tolerance to caffeine [25]. Consistently, the overexpression of a version of CaPpz1 carrying the same mutation restores normal caffeine tolerance but does not normalize growth in the presence of Li^+^ [37]. Myristoylation of Ppz1 has been found to be important for proper localization of the phosphatase to the plasma membrane ([38] and our own unpublished results) and for the role of the enzyme in methionine-triggered Mup1 endocytosis [38]. We observe here that overexpression of UmPpz1 in *S. cerevisiae* does not mimic the toxic effect of the endogenous enzyme, even if the enzyme is expressed at substantial levels. However, this lack of toxicity in UmPpz1 is no necessarily due to the absence of a terminal Gly because, although mutation of this residue decreases the toxicity to some extent (our unpublished results), the expression of other putatively myristoylable Ppz1 enzymes, such as DhPpz1 or CnPpz1, is not toxic for the cell [19,24]. Therefore, putting our results within the context of the literature, it appears that myristoylation of Ppz1 is necessary for some, but not all, the functions of the phosphatase, and that the actual protein expressed in *U. maydis* is not myristoylated.

We show here that deletion of Ppz1 in *U. maydis* does not affect tolerance to Li^+^ or Na^+^ cations, oxidants such as hydrogen peroxide or high temperature. In contrast, and similarly to budding yeast, mutant cells are sensitive to caffeine and calcofluor white. These results suggest that in *U. maydis* UmPpz1 is potentially involved in the CWI pathway but not in the maintenance of monovalent cation homeostasis. It has been postulated that in *S. cerevisiae*, the link between Ppz phosphatases and the Slt2/Mpk1 pathway is based on the regulatory role of the phosphatase on potassium uptake [8,10]. However, ionic homeostasis mechanisms have been shown to be different in *U. maydis* and *S. cerevisiae* [39]. Thus, in contrast to budding yeast, the internal Na^+^/K^+^ ratio in *U. maydis* can be > 2 without detrimental effects. In addition, *U. maydis* has two ENA ATPases, UmEna1 and UmEna2. UmEna1 is a typical ENA ATPase, although it shows low-Na^+^/K^+^ discrimination, whereas UmEna2 shows a different endosomal/plasma membrane distribution, and its function is still unclear [39]. In addition, it has been demonstrated that *U. maydis* escapes from G_2_ phase by activating the CWI pathway [34], which contrasts with the G_2_ cell cycle arrest that occurs in *S. cerevisiae* in response to CWI pathway activation [40,41]. Therefore, even if the phosphatases in both organisms are functionally related to the CWI, the underlying mechanisms are likely different.

*Saccharomyces cerevisiae* Hal3/Vhs3/Cab3 proteins can be taken as a prototype of what is found, almost without exception, in the genome of Saccharomycetes. In contrast, in other ascomycota classes, such as Eurotiomycetes, Dothideomycetes or Sordariomycetes, the protein is reduced to the PD domain and lacks the acidic tail. *U. maydis* Hal3 is an exceptionally large protein in which the PD domain is located at the very C-terminus and lacks the acidic tail. Our results show that UmPpz1 interacts much better with ScHal3 in vitro than with full-length UmHal3, the PD domain or the PD_ScCter version, suggesting that UmPpz1 retains the ability to be regulated by Hal3-like proteins. Indeed, ScHal3 is a powerful inhibitor of UmPpz1 phosphatase activity. In contrast, none of the versions of UmHal3 assayed had any inhibitory effect on ScPpz1 or UmPpz1. Therefore, despite the ability to interact up to some extent with ScPpz1 or UmPpz1, the *Ustilago* Hal3-like proteins are completely ineffective as phosphatase inhibitors in vitro. This behavior, also observed for the *C. neoformans* Hal3a and Hal3b proteins, provides additional support to the idea that in Hal3-like proteins, the structural determinants responsible for binding and inhibition of Ppz1 are different [24,42,43].

The absence of any effect on caffeine or calcofluor white when UmHal3 is overexpressed in *U. maydis* is in agreement with the failure of UmHal3 to inhibit UmPpz1 in vitro, and strongly suggests that in *U. maydis*, UmHal3 is not a regulatory subunit of the Ppz1 phosphatase. This raises the question of how the activity of this phosphatase is regulated in *U. maydis*. As suggested for *C. neoformans*, one possibility would be through phosphorylation [24]. Indeed, two conserved phosphorylated residues (S683 and S690) have been identified near the C-terminus in *S. cerevisiae* ([44,45] and our own unpublished data). The equivalent residues (S497 and S506) can be also phosphorylated in CnPpz1 [46]. Although, as far as we know, no phosphoproteomic data has been deposited for *U. maydis*, it is worth noting that the C-terminal tail of CnPpz1 and UmPpz1 is rather conserved, and the mentioned Ser residues are also found in the *Ustilago* phosphatase (Ser487 and Ser496). In fact, analysis of potential phosphorylatable sites using the NetPhosYeast 1.0 server [47] reveals that it is likely that these residues (score 0.918 and 0.772, respectively) will be phosphorylated.

Our results also show that UmHal3 acts as a fully functional PPCDC in *S. cerevisiae* and *E. coli*, and that this ability is retained by its PD domain, suggesting that all assayed version are able to form productive homotrimeric enzymes and that the very long N-terminal extension is not necessary for UmHal3 PPCDC function. Therefore, our data indicates that, as found for CnHal3b and CnHal3a in *C. neoformans*, the biological role of UmHal3 may be restricted to the participation in the biosynthesis of CoA, and, in this case, it would be the unique PPCDC encoded in the *U. maydis* genome. This would fit with our finding that UmHal3 is an essential gene. From these results, it could be concluded that UmHal3 would not be a moonlighting protein. It is worth noting, however, that the long N-terminal extension of UmHal3 contains a putative VPS9 domain. Vps9 is a guanine nucleotide exchange factor involved in vesicle-mediated vacuolar protein transport [48]. A similar combination of VPS9 domain and PD domain in a Hal3-like protein can be also found in other species of the order Ustilaginales, such as the maize pathogenic fungus *Sporisorium scitamineum* and the sugarcane pathogenic fungus *Sporisorium reilianum*, but not in other fungi, even in the closely-related order of Malasseziales. The function of this specifically conserved region is unknown, and we observed that its overexpression devoid of the PD domain does not affect growth or tolerance to certain stresses. Therefore, although it the hypothesis that the UmHal3-like protein may have the potential to perform as a moonlighting protein cannot be fully discarded, in this case, the non-PPCDC function would be different than in *S. cerevisiae* and *C. albicans*. In any case, it must be noted that UmHal3 is not likely the *bona fide* Vps9 in *Ustilago*, since a closer protein (UMAG_05369.1) is encoded in the genome of this fungus.

Our results demonstrate that, in contrast with what has been found in *C. albicans* and *A. fumigatus*, Ppz1 is not a virulence determinant in *U. maydis*. Since we recently showed that Ppz1 is not relevant for virulence in *C. neoformans*, it must be concluded that a role of Ppz1 phosphatases in fungal virulence is not a general feature, and must be investigated for each specific case. It is worth mentioning that both *U. maydis* and *C. neoformans* belongs to basidiomycetes, and it is possible that in this phylogenetic branch, the roles in virulence of Ppz1 phosphatases could be assumed by other phosphatases.

Recent crystallization and elucidation of the PD domain of the *C. neoformans* CnHal3b isoform [24] showed the presence of an extra α-helix in the CnHal3b structure not present in AtHal3 (nor in human PPCDC), due to the 17-residue sequence insertion from Val^86^ to Gly^102^. Similarly, in comparison with AtHal3 and human PPCDC, UmHal3 has exactly in this same region an even larger extra insert (residues 1256–1289, Appendix A). Because of the essential nature of PPCDC function, the presence of this structural feature in basidiomycete fungi might serve as basis for the development of selective drugs for human beings and antifungal treatments for maize, targeting Coenzyme A biosynthesis by interfering with PPCDC catalytic site or trimerization interface. Drugs targeting this step in the pathway have not yet been reported [49]. In this regard, the presence of subtle structural differences in the essential *U. maydis* protein kinase GSK3 compared to the human enzyme have been utilized for developing specific kinase inhibitors as potential antifungals [50].

## 4. Materials and Methods

### 4.1. Yeast Strains and Growth Conditions

*U. maydis* cells were grown on YPD (10 g L^−1^ yeast extract, 20 g L^−1^ peptone and 20 g L^−1^ dextrose) or Complete Medium (CM) at 28 °C, unless otherwise stated [51]. *S. cerevisiae* cells were grown at 28 °C in YPD medium or, when containing plasmids, in synthetic (SD) medium lacking the appropriate selection agents [52]. Sporulation of *S. cerevisiae* diploid strains was accomplished by transferring cells to a liquid medium containing 10 g L^−1^ potassium acetate, 1 g L^−1^ yeast extract and 0.5 g L^−1^ glucose (pH 7.2) for several days.

### 4.2. Generation of Fungal Deletion Mutant Strains

Deletions of the genes encoding *U. maydis* UmPpz1 and UmHal3 was made as follows. Deletion cassettes were generated using the Golden Gate cloning technology [53]. Briefly, upstream flank (UF) and downstream flank (DF) of hal3 (umag_05916) or ppz1 (umag_04827) were generated by PCR on FB1 genomic DNA using oligonucleotide combinations. To construct Hal3 deletion cassette the following oligos were used: HAL3-2/HAL3-3 (UF fragment, 1004 bp) and HAL3-7/HAL3-8 (DF, 889 bp). To construct the ppz1 deletion cassette the following oligos were used: PPZ1-2/PPZ1-3 (UF, 1068 bp) and PPZ1-4/PPZ1-5 (DF, 1085 bp). PCR products and pUMa1507 (storage vector I for simple knockout with HygR) were cut and ligated into pUMa1467 (destination vector) in a one-pot BsaI restriction/ligation reaction. To construct the different strains, transformation of *U. maydis* diploid strain FBD11 protoplasts with the indicated constructions was performed as described previously [54]. Integration of the disruption cassette into the corresponding loci was verified in each case by diagnostic PCR. *U. maydis* DNA isolation was performed as previously described [54].

Plant infections were performed with the maize cultivar Early Golden Bantam (Old Seeds, Madison, WI, USA) as described previously [55]. For the isolation of progeny and meiotic studies, spores from tumors were collected 10–14 days after inoculation and plated on plates containing 2% water agar. After 2 days, microcolonies (with about 200 cells) were removed and resuspended in YPD and dilutions plated onto YPD plates with and without hygromycin as selective agent.

Fungal strains used in this work are described in Table 1.

### 4.3. Recombinant DNA Techniques and Plasmid Construction

The *Escherichia coli* DH5α strain was used as plasmid DNA host and grown in Luria-Bertani (LB) medium broth at 37 °C supplemented, if needed, with ampicillin (100 μg/mL). Recombinant techniques, and bacterial and yeast transformations were carried out by standard methods.

To overexpress Um*hal3* and derivatives in *U. maydis*, DNA fragments carrying the desired sequences were amplified by PCR, and cloned into the pRU11 plasmid [56] under the control of *crg1* promoter as NdeI-EcoRI fragments. The oligonucleotide combinations were: UmHal3 full-length (HAL3-10/HAL3-13, 4300 bp), UmHal3 1-1186 (HAL3-10/HAL3-11, 3574 bp) and UmHal3 1187-1430 (HAL3-12/HAL3-13, 751 bp). These plasmids were inserted by homologous recombination into the *cbx* locus as described in [56].

For expression in *S. cerevisiae*, UmPpz1 was cloned into different vectors. Cloning into YCplac111 (low-copy vector) was as follows. Plasmids YCp111-CaPpz1 [20] was introduced into GM2163 (a *dam*^-^
*E. coli* strain). The CaPpz1 ORF was removed by digestion with XbaI and HindIII, and replaced with the UmPpz1 ORF, amplified with oligonucleotides UmPpz1_XbaI and UmPpz1_HindIII from *U. maydis* genomic DNA, to yield YCp-UmPpz1. Then, the *PPZ1* promoter followed by the UmPpz1 ORF was released by digestion with EcoRI and HindIII and introduced into YEplac181 (high-copy vector) linearized with the same restriction enzymes. A C-terminally HA tagged version of UmPpz1 was constructed by PCR amplification from YCp-UmPpz1 using oligonucleotides UmPpz1_XbaI and UmPpz1_HA_HindIII and cloning into XbaI/HindIII digested YCp-UmPpz1 to yield YCp-UmPpz1-HA. Constructs containing the short version (1-8Δ) of UmPpz1 (from now on UmPpz1-s) were made in the same way except that the UmPpz1 ORF was amplified with oligonucleotides UmPpz1_short_XbaI as forward primer.

An HA-tagged version of UmPpz1 expressed from plasmid pYES2 (2-micron, *GAL1* promoter, *URA3* marker) was made by PCR amplification of the tagged ORF, using YCp-UmPpz1-HA as template, with oligonucleotides UmPpz1_ATG_BamHI (or UmPpz1_short_ATG_BamHI, for UmPpz1-s) and UmPpz1_HA_EcoRI, following by cloning into the pYES2 vector at BamHI and EcoRI restriction sites. Similarly, the HA-tagged version of ScPpz1 was amplified from plasmid pCM190-PPZ1-HA (our laboratory, unpublished data) with oligonucleotides ScPpz1-HA-NotI and ScPpz1_in_PacI and cloned into pYES2 upon NotI and PacI digestion to yield pYES2-Ppz1-HA.

To overexpress UmHal3-like proteins with an N-terminal 3x-HA tag in *S. cerevisiae* and under the control of the *ADH1* promoter, the UmHal3 ORF (4995 bp) was amplified with oligonucleotides UmHal3_EcoRI and UmHal3_BamHI; then, it was inserted into the vector pWS93 [57] after digestion with EcoRI and BamHI. In addition, the C-terminal region (UmHal3_PD, corresponding to the putative PPCDC domain) was amplified with oligonucleotides UmHal3_PD_EcoRI and UmHal3_BamHI, and was then inserted into these sites of pWS93 as for UmHal3 ORF. The UmHal3_PDScCter version was obtained from plasmid pGEX-UmHal3_ScCter (see below) by digestion with EcoRI and XhoI and cloning into the plasmid pWS93 linearized by EcoRI and SalI (Appendix A).

For expression in *E. coli*, the UmPpz1 ORF was amplified from *U. maydis* FB1 genomic DNA with oligonucleotides UmPpz1_BamHI (UmPpz1_short_BamHI for UmPpz1-s) and UmPpz1_EcoRI. Then, the fragment was digested and inserted into the vector pGEX-6p-1 (GE Healthcare, Chicago, Il, USA) into the BamHI and EcoRI restriction sites. The UmHal3 and UmHal3_PD ORFs were amplified with the forward oligonucleotides UmHal3_EcoRI and UmHal3_PD_EcoRI, respectively, and the common reverse primer UmHal3_NotI. The fragments were then inserted into pGEX-6p-1 at the restriction sites EcoRI and NotI. The acidic tail of ScHal3 was added to UmHal3_PD by amplification of the UmHal3_PD sequence with oligonucleotides UmHal3_PD_EcoRI and UmHal3_ScCtD_RV using pWS-UmHal3 as template; the downstream C-terminal tail was generated with oligonucleotides UmHal3_ScCtD_FW and Reverse HAL3-XhoI using pWS-ScHal3 as template. The fusion fragment was amplified using UmHal3_PD_EcoRI and Reverse HAL3-XhoI and then inserted into pGEX-6p-1 by digestion with EcoRI and XhoI.

Oligonucleotide sequences can be found in Appendix A, and a list of plasmids used in this work is found in Appendix A.

### 4.4. Phenotypic Analysis

To test the sensitivity of *U. maydis* strains to different stress factors, cells were grown in YPD and ten-fold dilutions of this culture were spotted on CMD plates with the indicated chemicals. Cells were incubated for two days at 28 °C. For temperature sensitivity, CMD plates were incubated for two days at the indicated temperature. The sensitivity of *S. cerevisiae* cells was evaluated by spotting serial dilutions of starting cultures (OD_600_ of 0.05) on SD medium lacking uracil (for pWS93-based vectors) or leucine (for strains bearing YCplac111 or YEplac181 plasmids) and containing a range of concentrations of diverse compounds. Growth of strains overexpressing ScPpz1 or UmPpz1 (or their HA-tagged versions) from the *GAL1-10* promoter (pYES2-based vectors) was evaluated using synthetic medium lacking uracil and containing 2% galactose as carbon source. Plates were then incubated at 28 °C.

### 4.5. Expression and Purification of Recombinant Proteins

GST-ScPpz1, GST-ScPpz1-Cter, GST-UmPpz1, UmPpz1-s, GST-ScHal3, GST-UmHal3, GST-UmHal3_PD and GST-UmHal3_PD_ScCter were expressed using BL21 (DE3) RIL *E. coli* cells and purified as reported [58] with specific modifications [24]. The GST-tag was removed by PreScission protease (GE Healthcare), as described. The recombinant proteins were analyzed by SDS-PAGE followed by Coomassie Blue-staining and gels were scanned with an Epson Perfection V500 Photo scanner apparatus. Estimation of the amount of full length proteins was done with Gel Analyze 2010a software (http://www.gelanalyzer.com/) in comparison with bovine serum albumin (BSA) standards.

### 4.6. Yeasts Protein Extraction and Immunoblot Analysis

Yeast extracts were prepared as previously described [24,58]. Following SDS-PAGE, proteins were transferred to Immobilon-P membranes (Millipore, Burlington, MA, USA). and membranes were blocked and incubated with anti-HA antibodies (1:1000 dilution, Covance, Princeton, NJ, USA) followed by anti-mouse peroxidase antibodies (1:10000 dilution, GE Healthcare). Immunoreactive proteins were visualized using ECL western blotting detection kit (GE Healthcare) and chemiluminescence was detected in a Versadoc Image System 4000 MP (Bio-Rad Hercules, CA, USA).

### 4.7. In Vitro Interaction Experiments

The interaction between the Ppz phosphatases and the *U. maydis* proteins was determined by in vitro binding assays performed as previously described [58]. Aliquots of glutathione agarose beads containing 4 μg of the GST-tagged UmPpz1 (panel A) or 8 μg of ScPpz1 and ScPpz1-Cter were incubated with protein extracts (0.6–1.0 μg of proteins) prepared from strain IM021 (*ppz1Δ hal3Δ*) expressing the diverse 3HA-tagged Hal3 versions. The amount of protein extracts used was determined after detection and quantification by immunoblot of the expression levels of the relevant proteins (which were quite similar). The beads were finally resuspended in 100 μL of 2×DS sample buffer and, after boiling for 5 min, 25 or 50 μL of the samples were analyzed by SDS-PAGE and immunodetection.

### 4.8. Other Techniques

The *S. cerevisiae* and *U. maydis* Ppz1 phosphatase activity was assayed as previously described [58], using 5 to 10 pmol/assay and different amounts (0.5 to 4-fold molar excess) of the inhibitors. Phosphatase activity was measured using 10 mM of *p*-nitrophenylphosphate (Sigma Chem. Co, St. Louis, MO, USA) as substrate. The visible-UV absorption profile of recombinant UmHal3_PD devoid of the GST tag was obtained in a Nanodrop ND1000. Tetrad analysis were performed as in [15,27], respectively. Modeling of the PD region of UmHal3 was done at the Swiss-Model server (https://www.swissmodel.expasy.org/) [59].

## Figures and Tables

**Figure 1 ijms-20-03817-f001:**
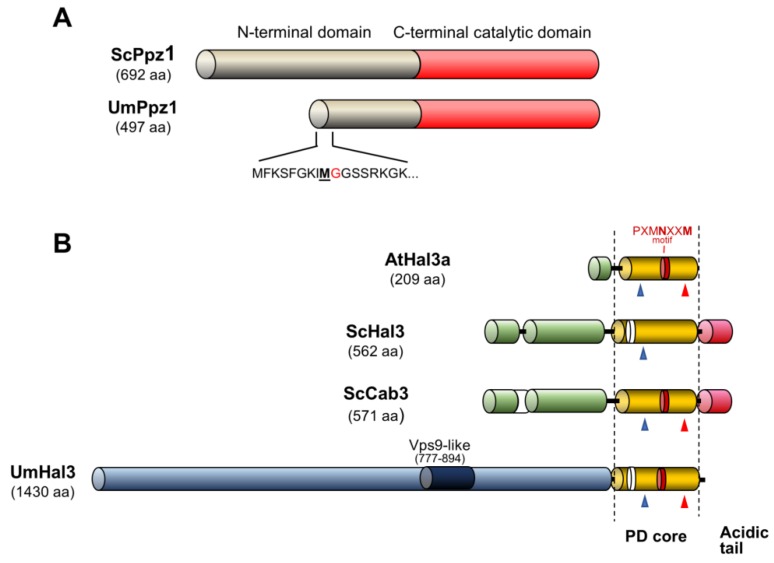
Comparison of the structures of *S. cerevisiae* and *U. maydis* Ppz1 and Hal3 proteins. Panel **A**. ScPpz1 and UmPpz1 structures. Note the shorter N-terminal extension in UmPpz1, the absence of a Gly at position 2, and the presence of a Met followed by a Gly (red) at positions 9 and 10. Panel **B**. Comparison of *A. thaliana* Hal3a, *S. cerevisiae* Hal3 and Cab3 and *U. maydis* Hal3 (UmHal3). The His and Cys residues required for PPCDC activity are indicated as blue and red arrowheads, respectively. The red section corresponds to the PXMNXXM motif. To note the absence in UmHal3 of the acidic C-tail found in both *S. cerevisiae* proteins. The length of each protein is indicated in parentheses. Scales in panel **A** and **B** are different.

**Figure 2 ijms-20-03817-f002:**
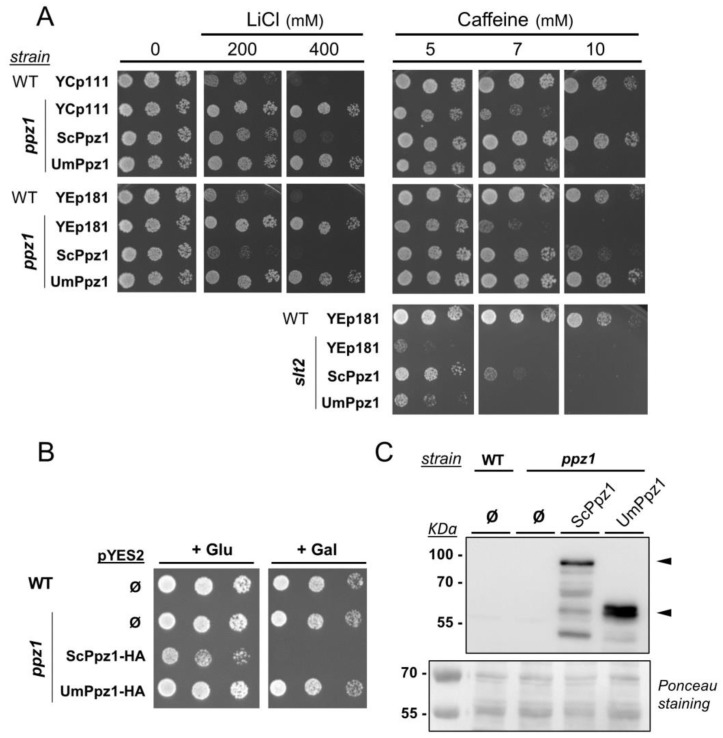
Phenotypic analysis of the expression of UmPpz1 in *S. cerevisiae*. (**A**) The wild type BY4741 strain (WT) and its *ppz1* and *slt2* derivatives were transformed with YCplac111 (centromeric, *LEU2* marker), or YEplac181 (episomal, *LEU2* marker) plasmids carrying *S. cerevisiae* (ScPpz1) or *U. maydis* (UmPpz1) phosphatases. Cultures were spotted at OD_660_=0.05 and at 1/5 dilutions on synthetic medium plates lacking leucine and containing the indicated amounts of LiCl or caffeine. Plates were grown for 4 days. Ø, empty plasmid. (**B**) Wild type and *ppz1* strains were transformed with empty pYES2 plasmid (pYES2-Ø) or the same plasmid expressing ScPpz1-HA or UmPpz1-HA from the *GAL1* promoter. Cells were grown on synthetic medium lacking uracil supplemented with 2% glucose and spotted on the same plates containing either 2% glucose or 2% galactose. Growth was monitored after 2 days. (**C**) The same strains were grown in liquid medium with 2% galactose. Samples were taken after 6 h and processed for SDS-polyacrylamide gel electrophoresis (SDS-PAGE) as described. 80 µg of protein were electrophoresed in 10% SDS-polyacrylamide gels, transferred to membranes and probed with anti-HA antibodies to detect the tagged proteins (arrowheads). Ponceau staining of the membranes was carried out (lower panel) to monitor for correct loading and transfer.

**Figure 3 ijms-20-03817-f003:**
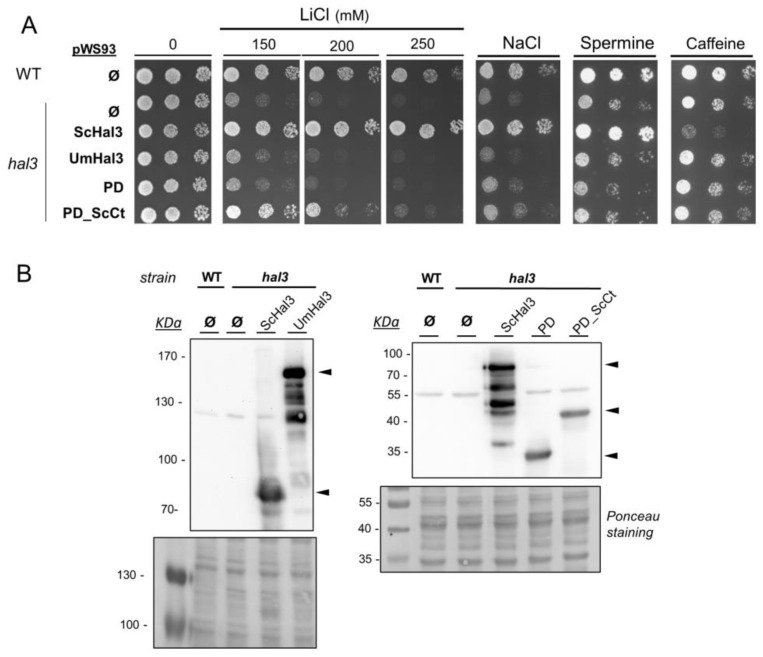
Effects of the expression of UmHal3 in *S. cerevisiae* on the tolerance to diverse stress conditions. (**A**) Wild type BY4741 (WT) and its *hal3* derivative were transformed with plasmid pWS93 (episomal, *URA3* marker) and the same plasmid expressing ScHal3, full-length UmHal3, its PPCDC domain (PD) and this domain followed by the acidic C-terminal tail form ScHal3 (PD_ScCt). Cells were spotted on synthetic medium plates lacking uracil supplemented with the indicated compounds. Growth was recorded after 48 h. (**B**) Protein extracts from cultures of the strains described in (**A**) were prepared, subjected to SDS-PAGE, and proteins were transferred to membranes. The different 3xHA-tagged proteins were detected with anti-HA antibodies (indicated by arrowheads). Ponceau staining of the membranes is shown.

**Figure 4 ijms-20-03817-f004:**
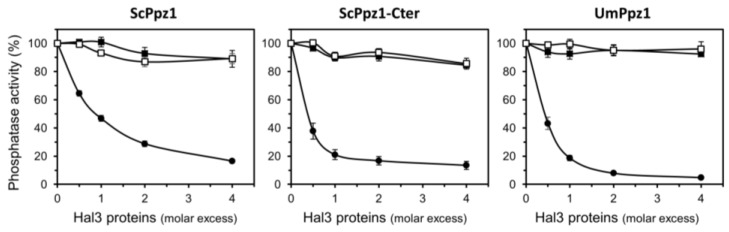
Inhibitory capacity of different UmHal3 versions on ScPpz1 and UmPpz1. Five to ten pmols of ScPpz1, ScPpz1-Cter, or UmPpz1 were pre-incubated for 5 min at 30 °C with different amounts of ScHal3 (-●-), UmHal3_PD (-■-) or UmHal3-ScCt (-☐-), and the assay was initiated by addition of the substrate. Data are means ± S.E. from 4 to 7 different assays and correspond to the ratio of phosphatase activity in the presence and the absence of inhibitor, expressed as percentage. Two different preparations of the phosphatases and the inhibitors were tested.

**Figure 5 ijms-20-03817-f005:**
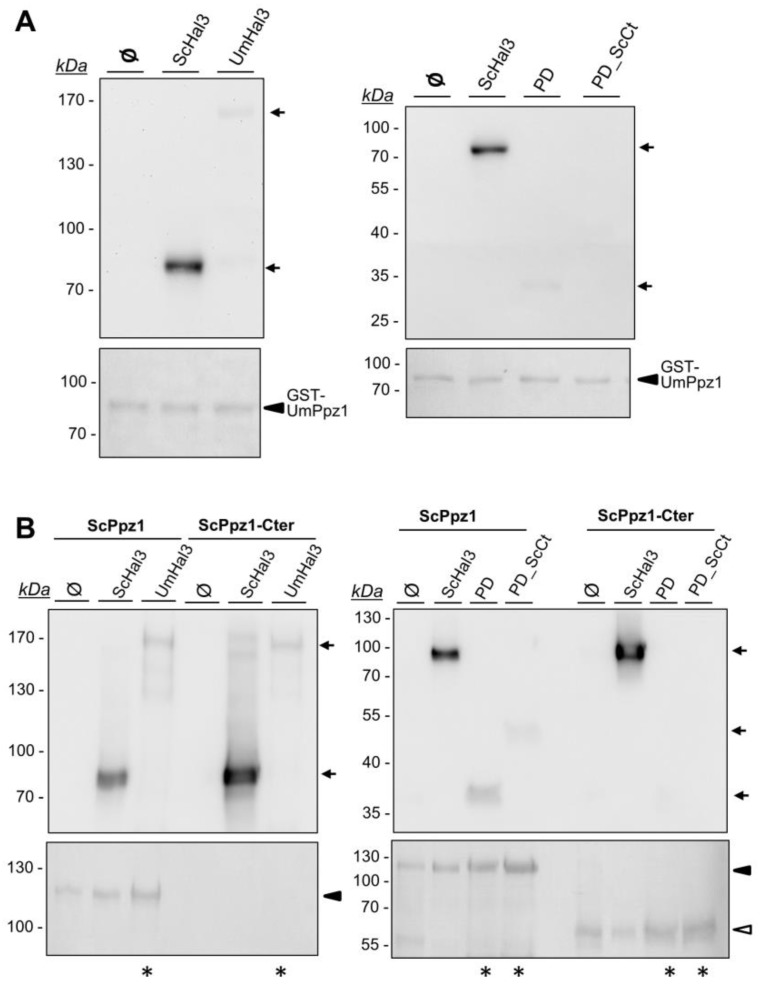
Evaluation of the interaction between UmHal3 variants with ScPpz1 and UmPpz1. Glutathione beads containing 4 μg of the GST-tagged UmPpz1 (panel **A**) or 8 μg of ScPpz1 and ScPpz1-Cter (panel **B**) were immobilized on glutathione beads and mixed with extracts from IM021 (ppz1 hal3) cells expressing 3HA-tagged ScHal3, UmHal3, UmHal3_PD (PD) or UmHal3_PD_ScCter (PD_ScCt) from the pWS93 plasmid. Beads were washed, resuspended in 100 μl of 2x sample buffer and processed for SDS-PAGE (6% gels for UmHal3 and 10% for PD and PD_ScCter). Typically, 25 μl of the samples was used, except in lanes marked with an asterisk, where 50 μl was employed. Proteins were immunoblotted using anti-HA antibodies as described in Methods. Ponceau staining of the membranes is shown to evaluate for correct loading and transfer (ScPpz1-Cter was not seen in the left blot in panel B because due to its relatively low molecular mass run with the front of the 6% gel).

**Figure 6 ijms-20-03817-f006:**
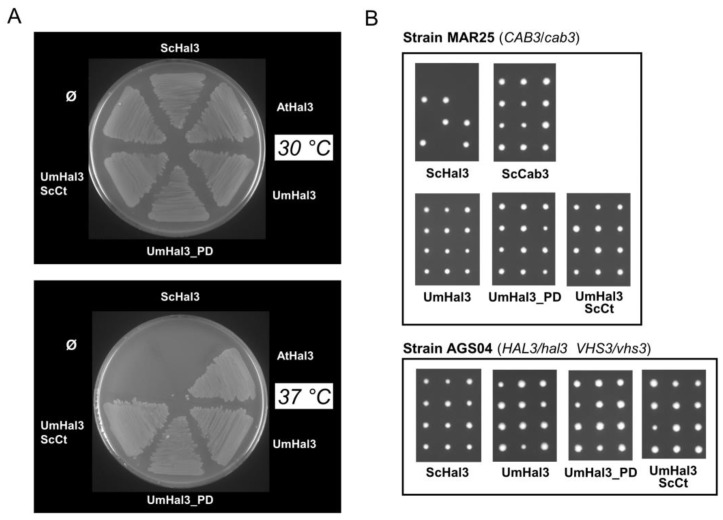
Evaluation of UmHal3 as functional PPCDC. (**A**) The empty pGEX6P-1 plasmid (∅) and the same vector bearing the *Arabidopsis* AtHal3 (as positive control), ScHal3 (as negative control), full-length UmHal3, and the indicated UmHal3 variants were introduced into the *E. coli* strain BW369, which carries the *dfp-707ts* mutation that abolish PPCDC activity when cells are grown at 37 °C. Transformants were grown at 30 °C, diluted till OD_600_ = 0.05, transferred to the permissive (30 °C) or 37 °C (non-permissive) temperatures for 6 h, and then plated at the indicated temperatures. Growth at 37 °C indicate complementation of the *ts* PPCDC mutation. (**B**) The heterozygous diploid strains MAR25 (*CAB3*/*cab3Δ*) and AGS04 (*HAL3*/*hal3Δ VHS3*/*vhs3Δ*) were transformed with the indicated plasmids, sporulation was induced, and spores from diverse asci (9 to 26 per transformation) were isolated with a micromanipulator and incubated for 3 days to form colonies. Note that *U. maydis* full-length UmHal3, as well as of its variants containing the PD domain, rescue the lethal phenotype of the *cab3* and *hal3 vhs3* mutations in haploidy, as denoted by germination and growth of all four spores in the tetrad.

**Figure 7 ijms-20-03817-f007:**
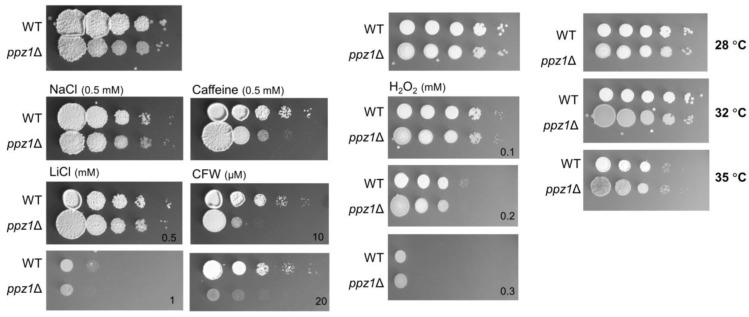
Phenotypic analysis of the *U. maydis* UmPpz1 mutant. Wild type (WT) and UmPpz1-deficient cells (*ppz1Δ*) were grown in YPD and ten-fold dilutions of this culture were spotted on CMD plates with the indicated chemicals. Cells were incubated for two days at 28 °C. For temperature sensitivity, CMD plates were incubated for two days at the indicated temperature.

**Figure 8 ijms-20-03817-f008:**
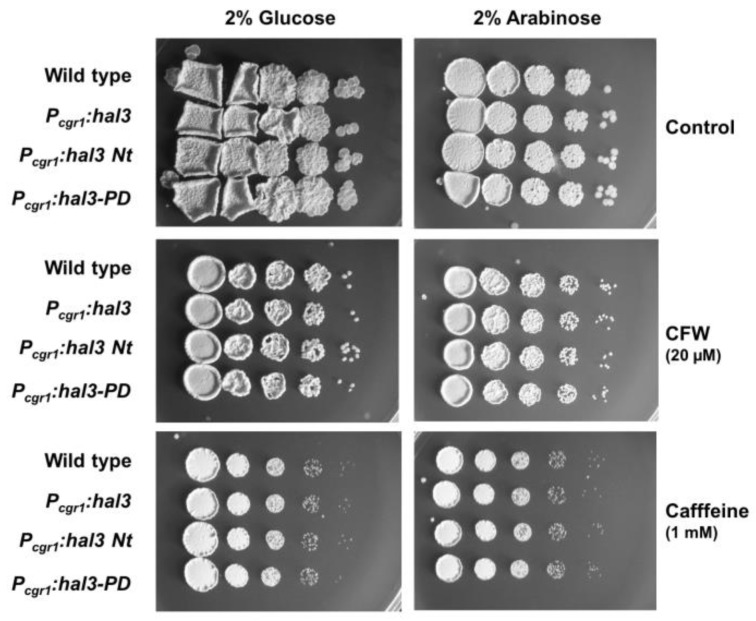
Growth *of U. maydis* strains overexpressing full-length UmHal3 and its N-terminal and C-terminal regions. Cells were grown in YPD and ten-fold dilutions of this culture were spotted on CMD (2% glucose, repressing conditions) or CMA (2% arabinose, de-repressing conditions) plates with the indicated chemicals. Pictures were taken after two days incubation at 28 °C.

**Figure 9 ijms-20-03817-f009:**
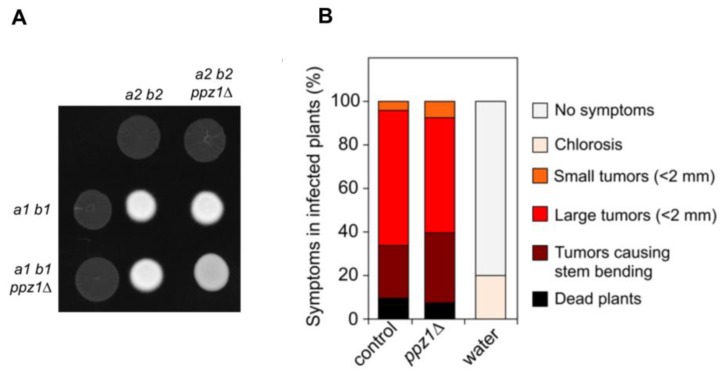
Analysis of virulence of mutants lacking Ppz1 function. (**A**) A *ppz1Δ* mutant is able to mate. Crosses of control as well as *ppz1Δ* mutant strains carrying compatible mating types (a1 b1 and a2 b2) in charcoal-containing agar plates. Positive fuzzy phenotype can be detected as a white-appearance mycelial growth. Note that mutant combinations were slightly affected in the ability to produce fuzzy phenotype. Plates were incubated at 22 °C for two days. (**B**) *ppz1Δ* mutant strains are able to infect plants. Disease symptoms caused by crosses of wild-type and *ppz1Δ* mutant strains. The symptoms were scored 14 days after infection. Two independent experiments were carried out and the average values are expressed as percentage of the total number of infected plants (n: 30 plants in each experiment).

**Table 1 ijms-20-03817-t001:** List of fungal strains used in this work and their genotypes.

Organism	Strain	Genotype	Source/Reference
*U. maydis*			
	FB1	*a1 b1*	[35]
	FB2	*a2 b2*	[35]
	FBD11	*a1a2 b1b2*	[35]
	UMP341	*a1a2 b1b2 hal3Δ::HygR/hal3*	This work
	UMP342	*a1a2 b1b2 ppz1Δ::HygR/ppz1*	This work
	UMP351	*a1 b1 ppz1Δ::HygR*	This work
	UMP352	*a2 b2 ppz1Δ::HygR*	This work
*S. cerevisiae*			
	BY4741	*MATa his3Δ*1 *leu2Δ*0 *met1*5*Δ*0 *ura3Δ*0	Euroscarf
	*ppz1*	BY4741 *ppz1::kanMx4*	Euroscarf
	*hal3*	BY4741 *hal3*::*kanMx4*	Euroscarf
	*slt2*	BY4741 *slt2*::*kanMx4*	Euroscarf
	IM21	JA100 (*MATa, ura3-52 leu2-3,112 trp1*-1 *his4 can-1r*) *ppz1*::*kanMx4 hal3*::*LEU2*	[42]
	AGS04	1788 (*a*/α, *ura3*-52 *leu2-*3,112 *trp1-1 his4 can-1r*) *VHS3*/*vhs3*::*kanMx4 HAL3*/*hal3*::*LEU2*	[15]
	MAR25	1788 (*a*/α, *ura3-52 leu2-3,112 trp1-1 his4 can-1r*) *CAB3*/*cab3*::*kanMx4*	[25]

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
