# Peer review of "Protein Phosphatase Ppz1 Is Not Regulated by a Hal3-Like Protein in Plant Pathogen Ustilago maydis"

_ijms, 2019, doi:10.3390/ijms20153817_

Round 1

Reviewer 1 Report

Major points

In introduction and results, the authors mention the second Gly of ScPpz1 is myristoylated and this is important. However, they did not mention the importance and function of this modification until the discussion. The description should be revised or it is very difficult to follow why the author want to mention these.

The author want to test myristoylation, so they deleted the first 8 amino acids of UmPpz1 but this resulted very low protein levels (line 155). From this result, the only conclusion can be obtained is the first 8 amino acids might be important for protein stability but no other conclusion could be drawn. If the myristoylation is important, the author can mutate the 2nd amino acid of UmPpz1 back to Gly and check whether this can change the properties of this enzyme. Alternatively, they can swap the very N terminal fragments between ScPpz1 and UmPpz1.

In line 169, the authors mentioned overexpression of ScPpz1 had harmful effects. What is the effect from? Cell wall integrity or ion imbalance? If the Gly2 is mutated to prevent Ppz1 to locate on the plasma membrane, is overexpression of ScPpz1 still detrimental? This result will help to compare between ScPpz1 and UmPpz1.

For in vitro interaction in Fig 5, it is necessary to include the GST-only as control. Some interaction bands are very weak and might be from the non-specific interaction signals. So it is important to include GST control to exclude this potential problem. Besides, the protein levels of 3HA‐tagged ScHal3, UmHal3, UmHal3_PD (PD) or UmHal3_PD_ScCter (PD_ScCt) should be included. This will help to evaluate the differences of interaction signals are from expression levels or interaction strength? One more thing, why the ScPpz1-Cter at left panel did not show any signals? If the controls above are not included, it is too early to conclude the proteins have direct physical interaction.

Minor points:

The description should be more clear and specific. For example, in line 48: …their terminal half…, does this mean the N terminus? Besides, there are some gramma errors in the articles should be corrected. In addition, some words need to be italicized. i.e. line 23. S. cerevisiae; line 24. in vitro and etc.

Author Response

Response to Referee #1

We appreciate very much the comments of the referee, which we feel help improving our paper.

Point 1. In introduction and results, the authors mention the second Gly of ScPpz1 is myristoylated and this is important. However, they did not mention the importance and function of this modification until the discussion. The description should be revised or it is very difficult to follow why the author want to mention these.

R: The referee is right in that no clue is given in the introduction about the role of myristoylation, so we have included the following sentence in the Introduction “In S. cerevisiae, change of Gly-2 to Ala results in lack of myristoylation and loss of complementation of salt tolerance [25].”

Point 2. The author want to test myristoylation, so they deleted the first 8 amino acids of UmPpz1 but this resulted very low protein levels (line 155). From this result, the only conclusion can be obtained is the first 8 amino acids might be important for protein stability but no other conclusion could be drawn. If the myristoylation is important, the author can mutate the 2nd amino acid of UmPpz1 back to Gly and check whether this can change the properties of this enzyme. Alternatively, they can swap the very N terminal fragments between ScPpz1 and UmPpz1.

R: Actually, we did not want to test for myristoylation, and we are sorry if our writing was somehow confusing in this respect. As we stated in the Results section with regard of the Gly in position 10 “computational analysis of the residues following this Gly does not predict this residue being myristoylated [31]”. However, because most putative fungal Ppz homologs do have a Gly as a second residue (very often in a myristoylable setting) our concern was that the actual UmPpz1 protein might start at the second Met (although we never though that, even in this case, this Gly could be myristoylated). We have reformulated this sentence for clarity.  In fact, the referee points here to the actual output of our test: that the first eight residues could be important for protein stability, a concept that was in our minds but was not explicit in the text. We have added a sentence at the end of section 2.2 to emphasize this view.

Point 3. In line 169, the authors mentioned overexpression of ScPpz1 had harmful effects. What is the effect from? Cell wall integrity or ion imbalance? If the Gly2 is mutated to prevent Ppz1 to locate on the plasma membrane, is overexpression of ScPpz1 still detrimental? This result will help to compare between ScPpz1 and UmPpz1.

R: The investigation of the molecular basis of Ppz1 toxicity is a current research line in our group, but the precise reasons for such toxicity are still elusive. Earlier work in laboratory demonstrated that overexpression of Ppz1 results in cell cycle blockage at the G1/s transition, with decreased expression of G1 cyclins (our reference 11). Our unpublished work suggest that it is not a problem of cell wall integrity or ion imbalance, and that mutation of Gly2 to Ala attenuates the growth defect, but do not restores normal growth. Please, note that part of this information is provided in the discussion section and used for comparison. See, for instance:  “However, this lack of toxicity in UmPpz1 is no necessarily due to the absence of a terminal Gly because, although mutation of this residue decreases somewhat toxicity (our unpublished results), expression of other putatively myristoylable Ppz1 enzymes, such as DhPpz1 or CnPpz1, is not toxic for the cell [19,24].”.

Point 4. For in vitro interaction in Fig 5, it is necessary to include the GST-only as control. Some interaction bands are very weak and might be from the nonspecific interaction signals. So it is important to include GST control to exclude this potential problem. Besides, the protein levels of 3HA‐tagged ScHal3, UmHal3, UmHal3_PD (PD) or UmHal3_PD_ScCter (PD_ScCt) should be included. This will help to evaluate the differences of interaction signals are from expression levels or interaction strength?

R: Our group has been carrying out in vitro interaction experiments as the ones described in this paper for more than 20 years. Initially, we included GST alone in our experiments (see references 9 and 15 for examples) just to test for nonspecific interaction signals. We never detected any signal, so we are not including this control anymore. It is true that some of the signals that we identify in fig 5 are quite faint, but we would like to stress that, for instance, PD_Cter does not give any detectable signal. Therefore, these lanes work as an internal negative control, since one might expect that, if interaction with the GST moiety would be responsible for unspecific background bands, they ought to be seen also in these lanes. Please, note also that the very short time provided by the journal for submission of the revised version (10 days) does not allow for reproduction of the experiments in figure 5 including a GST control.

On the other hand, we appreciate very much the suggestion of the referee about including data concerning the protein levels of HA-tagged proteins. These levels were evaluated each time new extracts were prepared and the amount of immunoreactive protein quantified in order to use equivalent amounts of tagged proteins in the interaction experiments. We have included examples of the relevant immunoblots as a new Supplementary Figure A3. We wish to stress that the tagged prey Hal3 variants were in vast excess, as deduced by monitoring the unbound pass-through material. Therefore, we are quite confident that variations in the material retained by ScPpz1 or UmPpz1 are not due to limitation in the prey proteins.

One more thing, why the ScPpz1-Cter at left panel did not show any signals?

R: We assume that the question refers to the Ponceau staining of the left blot in panel B. The reason for not seeing ScPpz1-Cter in the membrane was explained in the legend of the figure Ponceau staining of the membranes is shown to evaluate for correct loading and transfer (ScPpz1-Cter was not seen in the left blot in panel B because due to its relatively low molecular mass run with the front of the 6% gel).”

Minor points:

The description should be more clear and specific. For example, in line 48: … their terminal half…, does this mean the N terminus? Besides, there are some gramma errors in the articles should be corrected. In addition, some words need to be italicized. i.e. line 23. S. cerevisiae; line 24. in vitro and etc.

R: Thank you very much for pointing these errors. We have fully revised the paper to avoid mistakes as much as possible.

Reviewer 2 Report

Zhang et al. describe the identification, recombinant production and characterisation of the Ser/Thr protein phosphatase Ppz1 and the phosphopantothenoyl-cysteine decarboxylase Hal3 from Ustilago maydis. In other fungi, among them Saccharomyces cerevisiae, Ppz1 is involved in cation homeostasis and regulated by Hal3 acting as an inhibitory subunit. Via a number of experiments, the authors show conclusively that UmPpz1 is not involved in cation homeostasis in U. maydis. In addition, they demonstrate that U. maydis Hal3 (which is extremely large in comparison to the Hal3 proteins of other organisms) hardly interacts with Ppz1 (from U. maydis or other fungi). Consequently, it is not able to inhibit this phosphatase, although Hal3 from S. cerevisiae is able to inhibit U. maydis Ppz1 in vitro.

The present manuscript describes solid, interesting work, which is clearly worth publishing.

Minor points:

Line 15: replace „PPC decarboxylase“ by „phosphopantothenoyl cysteine decarboxylase”.

Lines 20 to 21: „In contrast to C. albicans and A. fumigatus, UmPpz1 is not a virulence determinant“. This sentence makes no sense. C. albicans and A. fumigatus are no virulence determinants too, but organisms. This sentence should be changed accordingly.

Lines 41 to 42: „… are sensitive to Na+ and Li+, whereas its overexpression confers tolerance to these toxic cations“. I would not refer to Na+ as a toxic cation. Omit “toxic” or add “high concentrations of”.

Line 48: Replace „terminal“ by N-terminal“.

Line 68: Replace „reminiscent to“ by „reminiscent of“.

Line 100: See lines 20 to 21.

Line 118: Replace „C-ter“ by „C-terminal“.

Line 165: Replace „Eighty“ by „80“.

Line 235: Replace „(even weakly)“ by „(although weakly)“ or „(even though weakly)“.

Line 419: Replace „Vsp9“ by „Vps9“.

Author Response

Response to Referee #2

We appreciate very much the positive comments of the referee about our work and his /her suggestions for wording improvement, which we have closely followed.

Minor points:

Line 15: replace „PPC decarboxylase“ by „phosphopantothenoyl cysteine decarboxylase”.

R: Replaced

Lines 20 to 21: „In contrast to C. albicans and A. fumigatus, UmPpz1 is not a virulence determinant“. This sentence makes no sense. C. albicans and A. fumigatus are no virulence determinants too, but organisms. This sentence should be changed accordingly.

R: Very true. We were trying to compact the abstract not to overflow the word limit and this led to this ill-constructed sentence. It has been replaced by “In contrast to what was found in C. albicans and A. fumigatus, UmPpz1 is not a virulence determinant.”

Lines 41 to 42: „… are sensitive to Na+ and Li+, whereas its overexpression confers tolerance to these toxic cations“. I would not refer to Na+ as a toxic cation. Omit “toxic” or add “high concentrations of”.

R: Indeed, toxicity is a matter of dose, and in this sense, sodium is clearly less harmful than lithium. So, to make the sentence straight, now it is read “….. whereas its overexpression confers tolerance to high concentrations of these cations [13].”

Line 48: Replace „terminal“ by N-terminal“.

R: Replaced, sorry for the oversight.

Line 68: Replace „reminiscent to“ by „reminiscent of“.

R: Replaced.

Line 100: See lines 20 to 21.

R: Replaced by “In contrast to what has been reported for the phosphatase in C. albicans and A. fumigatus, UmPpz1 is not a virulence determinant.“

Line 118: Replace „C-ter“ by „C-terminal“.

R: Done.

Line 165: Replace „Eighty“ by „80“.

R: Done

Line 235: Replace „(even weakly)“ by „(although weakly)“ or „(even though weakly)“.

R: Replaced.

Line 419: Replace „Vsp9“ by „Vps9“.

R: Replaced.

Round 2

Reviewer 1 Report

 Accept in present form